# FEW-SHOT AUTOREGRESSIVE DENSITY ESTIMATION:
## TOWARDS LEARNING TO LEARN DISTRIBUTIONS

**S. Reed, Y. Chen, T. Paine, A. van den Oord, S. M. A. Eslami, D. Rezende, O. Vinyals, N. de Freitas**
{reedscot,yutianc,tpaine}@google.com

## ABSTRACT

Deep autoregressive models have shown state-of-the-art performance in density estimation for natural images on large-scale datasets such as ImageNet. However, such models require many thousands of gradient-based weight updates and unique image examples for training. Ideally, the models would rapidly learn visual concepts from only a handful of examples, similar to the manner in which humans learns across many vision tasks. In this paper, we show how 1) neural attention and 2) meta learning techniques can be used in combination with autoregressive models to enable effective few-shot density estimation. Our proposed modifications to PixelCNN result in state-of-the art few-shot density estimation on the Omniglot dataset. Furthermore, we visualize the learned attention policy and find that it learns intuitive algorithms for simple tasks such as image mirroring on ImageNet and handwriting on Omniglot without supervision. Finally, we extend the model to natural images and demonstrate few-shot image generation on the Stanford Online Products dataset.

## 1 INTRODUCTION

Contemporary machine learning systems are still far behind humans in their ability to rapidly learn new visual concepts from only a few examples (Lake et al., 2013). This setting, called few-shot learning, has been studied using deep neural networks and many other approaches in the context of discriminative models, for example Vinyals et al. (2016); Santoro et al. (2016). However, comparatively little attention has been devoted to the task of few-shot image density estimation; that is, the problem of learning a model of a probability distribution from a small number of examples. Below we motivate our study of few-shot autoregressive models, their connection to meta-learning, and provide a comparison of multiple approaches to conditioning in neural density models.

### WHY AUTOREGRESSIVE MODELS?

Autoregressive neural networks are useful for studying few-shot density estimation for several reasons. They are fast and stable to train, easy to implement, and have tractable likelihoods, allowing us to quantitatively compare a large number of model variants in an objective manner. Therefore we can easily add complexity in orthogonal directions to the generative model itself.

Autoregressive image models factorize the joint distribution into per-pixel factors:

$$P(\mathbf{x}|s; \theta) = \prod_{t=1}^{N} P(x_t|x_{<t}, f(s); \theta) \tag{1}$$

where $\theta$ are the model parameters, $x \in \mathbb{R}^N$ are the image pixels, $s$ is a conditioning variable, and $f$ is a function encoding this conditioning variable. For example in text-to-image synthesis, $s$ would be an image caption and $f$ could be a convolutional or recurrent encoder network, as in Reed et al. (2016). In label-conditional image generation, $s$ would be the discrete class label and $f$ could simply convert $s$ to a one-hot encoding possibly followed by an MLP.

A straightforward approach to few-shot density estimation would be to simply treat samples from the target distribution as conditioning variables for the model. That is, let $s$ correspond to a few data examples illustrating a concept. For example, $s$ may consist of four images depicting bears, and the task is then to generate an image $x$ of a bear, or to compute its probability $P(x|s; \theta)$.

A learned conditional density model that conditions on samples from its target distribution is in fact learning a learning algorithm, embedded into the weights of the network. This learning algorithm is executed by a feed-forward pass through the network encoding the target distribution samples.

### WHY LEARN TO LEARN DISTRIBUTIONS?

If the number of training samples from a target distribution is tiny, then using standard gradient descent to train a deep network from scratch or even fine-tuning is likely to result in memorization of the samples; there is little reason to expect generalization. Therefore what is needed is a learning algorithm that can be expected to work on tiny training sets. Since designing such an algorithm has thus far proven to be challenging, one could try to learn the algorithm itself. In general this may be impossible, but if there is shared underlying structure among the set of target distributions, this learning algorithm can be learned from experience as we show in this paper.

For our purposes, it is instructive to think of learning to learn as two nested learning problems, where the inner learning problem is less constrained than the outer one. For example, the inner learning problem may be unsupervised while the outer one may be supervised. Similarly, the inner learning problem may involve only a few data points. In this latter case, the aim is to meta-learn a model that when deployed is able to infer, generate or learn rapidly using few data $s$.

A rough analogy can be made to evolution: a slow and expensive meta-learning process, which has resulted in life-forms that at birth already have priors that facilitate rapid learning and inductive leaps. Understanding the exact form of the priors is an active, very challenging, area of research (Spelke & Kinzler, 2007; Smith & Gasser, 2005). From this research perspective, we can think of meta-learning as a potential data-driven alternative to hand engineering priors.

The meta-learning process can be undertaken using large amounts of computation and data. The output is however a model that can learn from few data. This facilitates the deployment of models in resource-constrained computing devices, e.g. mobile phones, to learn from few data. This may prove to be very important for protection of private data $s$ and for personalisation.

### FEW-SHOT LEARNING AS INFERENCE OR AS A WEIGHT UPDATE?

A sample-conditional density model $P_\theta(x|s)$ treats meta-learning as inference; the conditioning samples $s$ vary but the model parameters $\theta$ are fixed. A standard MLP or convolutional network can parameterize the sample encoding (i.e. meta-learning) component, or an attention mechanism can be used, which we will refer to as PixelCNN and Attention PixelCNN, respectively.

A very different approach to meta-learning is taken by Ravi & Larochelle (2016) and Finn et al. (2017a), who instead learn *unconditional* models that adapt their weights based on a gradient step computed on the few-shot samples. This same approach can also be taken with PixelCNN: train an unconditional network $P_{\theta'}(x)$ that is implicitly conditioned by a previous gradient ascent step on $\log P_\theta(s)$; that is, $\theta' = \theta - \alpha \nabla_\theta \log P_\theta(s)$. We will refer to this as Meta PixelCNN.

In Section 2 we connect our work to previous attentive autoregressive models, as well as to work on gradient based meta-learning. In Section 3 we describe Attention PixelCNN and Meta PixelCNN in greater detail. We show how attention can improve performance in the the few-shot density estimation problem by enabling the model to easily transmit texture information from the support set onto the target image canvas. In Section 4 we compare several few-shot PixelCNN variants on simple image mirroring, Omniglot and Stanford Online Products. We show that both gradient-based and attention-based few-shot PixelCNN can learn to learn simple distributions, and both achieve state-of-the-art likelihoods on Omniglot.

## 2 RELATED WORK

Learning to learn or meta-learning has been studied in cognitive science and machine learning for decades (Harlow, 1949; Thrun & Pratt, 1998; Hochreiter et al., 2001). In the context of modern deep networks, Andrychowicz et al. (2016) learned a gradient descent optimizer by gradient descent, itself parameterized as a recurrent network. Chen et al. (2017) showed how to learn to learn by gradient descent in the black-box optimization setting.

Ravi & Larochelle (2017) showed the effectiveness of learning an optimizer in the few-shot learning setting. Finn et al. (2017a) advanced a simplified yet effective variation in which the optimizer is not learned but rather fixed as one or a few steps of gradient descent, and the meta-learning problem reduces to learning an initial set of base parameters $\theta$ that can be adapted to minimize any task loss $\mathcal{L}_t$ by a single step of gradient descent, i.e. $\theta' = \theta - \alpha \nabla \mathcal{L}_t(\theta)$. This approach was further shown to be effective in imitation learning including on real robotic manipulation tasks (Finn et al., 2017b). Shyam et al. (2017) train a neural attentive recurrent comparator function to perform one-shot classification on Omniglot.

Few-shot density estimation has been studied previously using matching networks (Bartunov & Vetrov, 2016) and variational autoencoders (VAEs). Bornschein et al. (2017) apply variational inference to memory addressing, treating the memory address as a latent variable. Rezende et al. (2016) develop a sequential generative model for few-shot learning, generalizing the Deep Recurrent Attention Writer (DRAW) model (Gregor et al., 2015). In this work, our focus is on extending autoregressive models to the few-shot setting, in particular PixelCNN (van den Oord et al., 2016).

Autoregressive (over time) models with attention are well-established in language tasks. Bahdanau et al. (2014) developed an attention-based network for machine translation. This work inspired a wave of recurrent attention models for other applications. Xu et al. (2015) used *visual* attention to produce higher-quality and more interpretable image captioning systems. This type of model has also been applied in motor control, for the purpose of imitation learning. Duan et al. (2017) learn a policy for robotic block stacking conditioned on a small number of demonstration trajectories.

Gehring et al. (2017) developed convolutional machine translation models augmented with attention over the input sentence. A nice property of this model is that all attention operations can be *batched* over time, because one does not need to unroll a recurrent net during training. Our attentive Pixel-CNN is similar in high-level design, but our data is pixels rather than words, and 2D instead of 1D, and we consider image generation rather than text generation as our task.

## 3 MODEL

### 3.1 FEW-SHOT LEARNING WITH ATTENTION PIXELCNN

In this section we describe the model, which we refer to as Attention PixelCNN. At a high level, it works as follows: at the point of generating every pixel, the network queries a memory. This memory can consist of anything, but in this work it will be a support set of images of a visual concept. In addition to global features derived from these support images, the network has access to textures via support image patches. Figure 2 illustrates the attention mechanism.

In previous conditional PixelCNN works, the encoding $f(s)$ was shared across all pixels. However, this can be sub-optimal for several reasons. First, at different points of generating the target image $x$, different aspects of the support images may become relevant. Second, it can make learning difficult, because the network will need to encode the entire support set of images into a single global conditioning vector, fed to every output pixel. This single vector would need to transmit information across all pairs of salient regions in the supporting images and the target image.

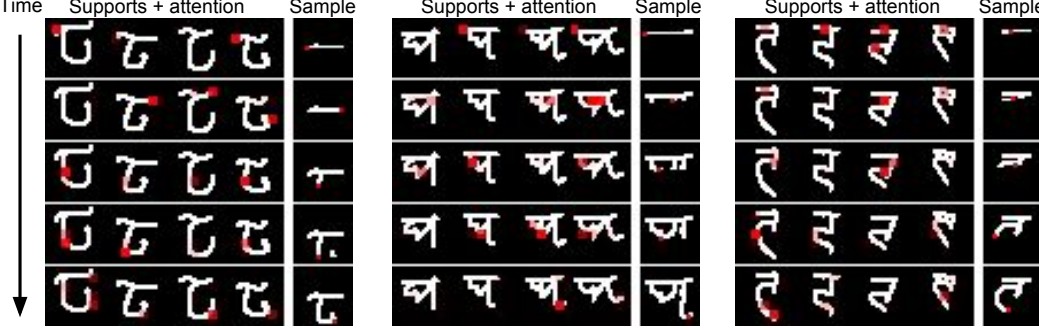

Figure 1: Sampling from Attention PixelCNN. Support images are overlaid in red to indicate the attention weights. The support sets can be viewed as small training sets, illustrating the connection between sample-conditional density estimation and learning to learn distributions.

To overcome this difficulty, we propose to replace the simple encoder function $f(s)$ with a context-sensitive attention mechanism $f_t(s, x_{<t})$. It produces an encoding of the context that depends on the image generated up until the current step $t$. The weights are shared over $t$.

We will use the following notation. Let the *target image* be $x \in \mathbb{R}^{H \times W \times 3}$. and the support set images be $s \in \mathbb{R}^{S \times H \times W \times 3}$, where $S$ is the number of supports.

To capture texture information, we encode all supporting images with a shallow convolutional network, typically only two layers. Each hidden unit of the resulting feature map will have a small receptive field, e.g. corresponding to a $10 \times 10$ patch in a support set image. We encode these support images into a set of spatially-indexed key and value vectors.

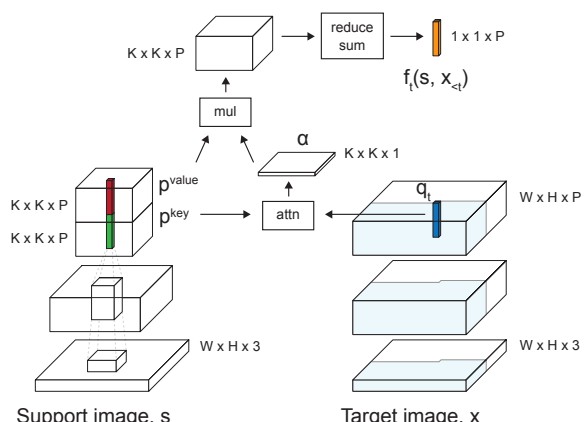

Figure 2: The PixelCNN attention mechanism.

After encoding the support images in parallel, we reshape the resulting $S \times K \times K \times 2P$ feature maps to squeeze out the spatial dimensions, resulting in a $SK^2 \times 2P$ matrix.

$$p = f_{patch}(s) = \text{reshape}(\text{CNN}(s), [SK^2 \times 2P]) \tag{2}$$

$$p^{key} = p[:, 0:P], \; p^{value} = p[:, P:2P] \tag{3}$$

where CNN is a shallow convolutional network. We take the first $P$ channels as the patch key vectors $p^{key} \in \mathbb{R}^{SK^2 \times P}$ and the second $P$ channels as the patch value vectors $p^{value} \in \mathbb{R}^{SK^2 \times P}$. Together these form a queryable memory for image generation.

To query this memory, we need to encode both the global context from the support set $s$ as well as the pixels $x_{<t}$ generated so far. We can obtain these features simply by taking any layer of a PixelCNN conditioned on the support set:

$$q_t = \text{PixelCNN}_L(f(s), x_{<t}), \tag{4}$$

where $L$ is the desired layer of hidden unit activations within the PixelCNN network. In practice we use the middle layer.

To incorporate the patch attention features into the pixel predictions, we build a scoring function using $q$ and $p^{key}$. Following the design proposed by Bahdanau et al. (2014), we compute a normalized matching score $\alpha_{tj}$ between query pixel $q_t$ and supporting patch $p_j^{key}$ as follows:

$$e_{tj} = v^T \tanh(q_t + p_j^{key}) \tag{5}$$

$$\alpha_{tj} = \exp(e_{tj}) / \sum_{k=1}^{SK^2} \exp(e_{ik}). \tag{6}$$

The resulting attention-gated context function can be written as:

$$f_t(s, x_{<t}) = \sum_{j=1}^{SK^2} \alpha_{tj} p_j^{value} \tag{7}$$

which can be substituted into the objective in equation 1. In practice we combine the attention context features $f_t(s, x_{<t})$ with global context features $f(s)$ by channel-wise concatenation.

This attention mechanism can also be straightforwardly applied to the multiscale PixelCNN architecture of Reed et al. (2017). In that model, pixel factors $P(x_t|x_{<t}, f_t(s, x_{<t}))$ are simply replaced by pixel group factors $P(x_g|x_{<g}, f_g(s, x_{<g}))$, where $g$ indexes a set of pixels and $< g$ indicates all pixels in previous pixel groups, including previously-generated lower resolutions.

We find that a few simple modifications to the above design can significantly improve performance. First, we can augment the supporting images with a channel encoding relative position within the image, normalized to $[-1, 1]$. One channel is added for $x$-position, another for $y$-position. When

patch features are extracted, position information is thus encoded, which may help the network assemble the output image. Second, we add a 1-of-$K$ channel for the supporting image label, where $K$ is the number of supporting images. This provides patch encodings information about which global context they are extracted from, which may be useful e.g. when assembling patches from multiple views of an object.

## 3.2 Few-shot learning with Meta PixelCNN

As an alternative to explicit conditioning with attention, in this section we propose an implicitly-conditioned version using gradient descent. This is an instance of what Finn et al. (2017a) called *model-agnostic* meta learning, because it works in the same way regardless of the network architecture. The conditioning pathway (i.e. flow of information from supports $s$ to the next pixel $x_t$) introduces no additional parameters. The objective to maximize is as follows:

$$\mathcal{L}(x, s; \theta) = \log P(x; \theta'), \text{ where } \theta' = \theta - \alpha \nabla_\theta \mathcal{L}_{\text{inner}}(s; \theta) \tag{8}$$

A natural choice for the inner objective would be $\mathcal{L}_{\text{inner}}(s; \theta) = \log P(s; \theta)$. However, as shown in Finn et al. (2017b) and similar to the setup in Neu & Szepesvári (2012), we actually have considerable flexibility here to make the inner and outer objectives different.

Any learnable function of $s$ and $\theta$ could potentially learn to produce gradients that increase $\log P(x; \theta')$. In particular, this function does not need to compute log likelihood, and does not even need to respect the causal ordering of pixels implied by the chain rule factorization in equation 1. Effectively, the model can learn to learn by maximum likelihood without likelihoods.

As input features for computing $\mathcal{L}_{\text{inner}}(s, \theta)$, we use the $L$-th layer of spatial features $q = \text{PixelCNN}_L(s) \in \mathbb{R}^{H \times W \times Z}$, where $Z$ is the number of feature channels used in the PixelCNN. Note that this is the same network used to model $P(x; \theta)$. The features $q$ are fed through a convolutional network producing a scalar, which is the (learned) inner loss. In practice, we used $\alpha = 0.1$, and the encoder had three layers of stride-2 convolutions with $3 \times 3$ kernels, followed by element-wise squaring and a sum over all dimensions. Since these convolutional weights are part of $\theta$, they are learned jointly with the generative model weights by maximizing equation 8.

## 4 Experiments

In this section we describe experiments on image flipping, Omniglot, and Stanford Online Products. In all experiments, the support set encoder $f(s)$ has the following structure: in parallel over support images, a $5 \times 5$ conv layer, followed by a sequence of $3 \times 3$ convolutions and max-pooling until the spatial dimension is $1$. Finally, the support image encodings are concatenated and fed through two fully-connected layers to get the support set embedding.

### 4.1 ImageNet Flipping

As a diagnostic task, we consider the problem of image flipping as few-shot learning. The "support set" contains only one image and is simply the horizontally-flipped target image. A trivial algorithm exists for this problem, which of course is to simply copy pixel values directly from the support to the corresponding target location. We find that the Attention PixelCNN did indeed learn to solve the task, however, interestingly, the baseline conditional PixelCNN and Meta PixelCNN did not.

We trained the model on ImageNet (Deng et al., 2009) images resized to $48 \times 48$ for $30K$ steps using RMSProp with learning rate $1e^{-4}$. The network was a 16-layer PixelCNN with 128-dimensional feature maps at each layer, with skip connections to a 256-dimensional penultimate layer before pixel prediction. The baseline PixelCNN is conditioned on the 128-dimensional encoding of the flipped image at each layer; $f(s) = f(x')$, where $x'$ is the mirror image of $x$. The Attention PixelCNN network is exactly the same for the first $8$ layers, and the latter $8$ layers are conditioned also on attention features $f_t(s, x_{<t}) = f_t(x', x_{<t})$ as described in section 3.1.

Figure 3 shows the qualitative results for several validation set images. We observe that the baseline model without attention completely fails to flip the image or even produce a similar image. With attention, the model learns to consistently apply the horizontal flip operation. However, it is not

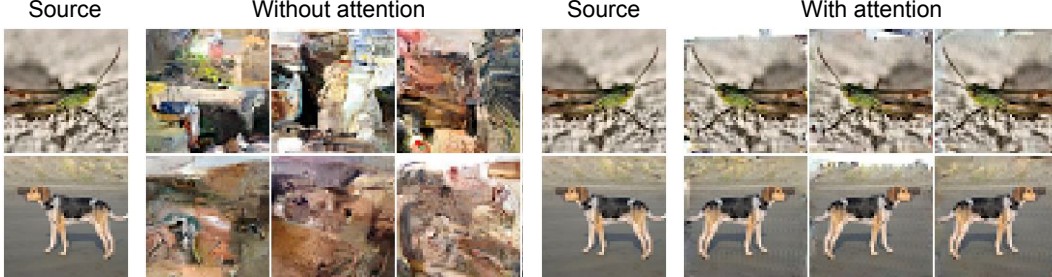

Figure 3: Horizontally flipping ImageNet images. The network using attention learns to mirror, while the network without attention does not.

entirely perfect - one can observe slight mistakes on the upper and left borders. This makes sense because in those regions, the model has the least context to predict pixel values. We also ran the experiment on $24 \times 24$ images; see figure 6 in the appendix. Even in this simplified setting, neither the baseline conditional PixelCNN or Meta PixelCNN learned to flip the image.

Quantitatively, we also observe a clear difference between the baseline and the attention model. The baseline achieves $2.64$ nats/dim on the training set and $2.65$ on the validation set. The attention model achieves $0.89$ and $0.90$ nats/dim, respectively. During sampling, Attention PixelCNN learns a simple copy operation in which the attention head proceeds in right-to-left raster order over the input, while the output is written in left-to-right raster order.

## 4.2 OMNIGLOT

In this section we benchmark our model on Omniglot (Lake et al., 2013), and analyze the learned behavior of the attention module. We trained the model on $26 \times 26$ binarized images and a $45 - 5$ split into training and testing character alphabets as in Bornschein et al. (2017).

To avoid over-fitting, we used a very small network architecture. It had a total of $12$ layers with $24$ planes each, with skip connections to a penultimate layer with $32$ planes. As before, the baseline model conditioned each pixel prediction on a single global vector computed from the support set. The attention model is the same for the first half (6 layers), and for the second half it also conditions on attention features.

The task is set up as follows: the network sees several images of a character from the same alphabet, and then tries to induce a density model of that character. We evaluate the likelihood on a held-out example image of that same character from the same alphabet.

| Model | Number of support set examples | | | |
|---|---|---|---|---|
| | **1** | **2** | **4** | **8** |
| Bornschein et al. (2017) | 0.128(−−) | 0.123(−−) | 0.117(−−) | − − (−−) |
| Gregor et al. (2016) | 0.079(0.063) | 0.076(0.060) | 0.076(0.060) | 0.076(0.057) |
| Conditional PixelCNN | 0.077(0.070) | 0.077(0.068) | 0.077(0.067) | 0.076(0.065) |
| Attention PixelCNN | **0.071(0.066)** | **0.068(0.064)** | **0.066(0.062)** | **0.064(0.060)** |

Table 1: Omniglot test(train) few-shot density estimation NLL in nats/dim. Bornschein et al. (2017) refers to Variational Memory Addressing and Gregor et al. (2016) to ConvDRAW.

All PixelCNN variants achieve state-of-the-art likelihood results (see table 1). Attention PixelCNN significantly outperforms the other methods, including PixelCNN without attention, across $1, 2, 4$ and $8$-shot learning. PixelCNN and Attention PixelCNN models are also fast to train: $10K$ iterations with batch size 32 took under an hour using NVidia Tesla K80 GPUs.

We also report new results of training a ConvDRAW Gregor et al. (2016) on this task. While the likelihoods are significantly worse than those of Attention PixelCNN, they are otherwise state-of-the-art, and qualitatively the samples look as good. We include ConvDRAW samples on Omniglot for comparison in the appendix section 6.2.

| PixelCNN Model | NLL test(train) |
|---|---|
| Conditional PixelCNN | 0.077(0.067) |
| Attention PixelCNN | 0.066(0.062) |
| Meta PixelCNN | 0.068(0.065) |
| Attention Meta PixelCNN | 0.069(0.065) |

Table 2: Omniglot NLL in nats/pixel with four support examples. Attention Meta PixelCNN is a model combining attention with gradient-based weight updates for few-shot learning.

Meta PixelCNN also achieves state-of-the-art likelihoods, only outperformed by Attention Pixel-CNN (see Table 2). Naively combining attention and meta learning does not seem to help. However, there are likely more effective ways to combine attention and meta learning, such as varying the inner loss function or using multiple meta-gradient steps, which could be future work.

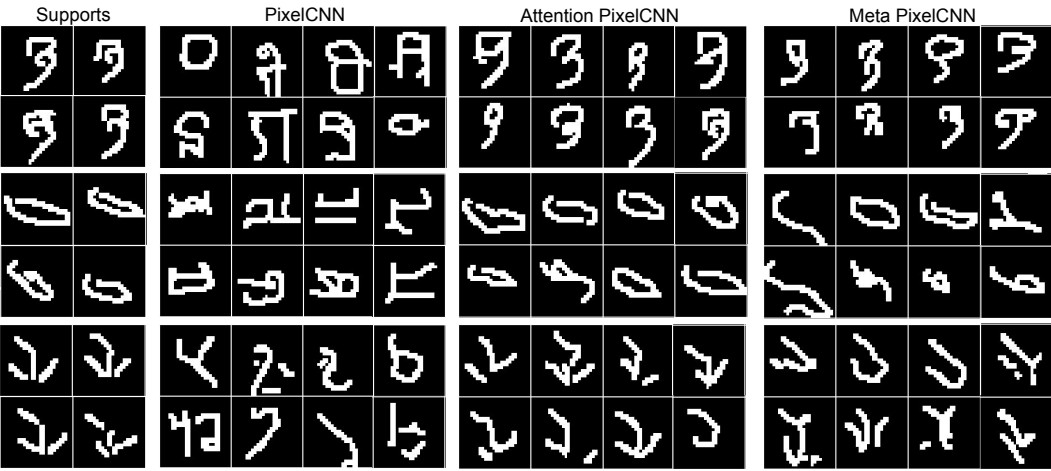

Figure 4: Typical Omniglot samples from PixelCNN, Attention PixelCNN, and Meta PixelCNN.

Figure 1 shows several key frames of the attention model sampling Omniglot. Within each column, the left part shows the 4 support set images. The red overlay indicates the attention head read weights. The red attention pixel is shown over the center of the corresponding patch to which it attends. The right part shows the progress of sampling the image, which proceeds in raster order. We observe that as expected, the network learns to attend to corresponding regions of the support set when drawing each portion of the output image. Figure 4 compares results with and without attention. Here, the difference in likelihood clearly correlates with improvement in sample quality.

## 4.3 STANFORD ONLINE PRODUCTS

In this section we demonstrate results on natural images from online product listings in the Stanford Online Products Dataset (Song et al., 2016). The data consists of sets of images showing the same product gathered from eBay product listings. There are 12 broad product categories. The training set has 11, 318 distinct objects and the testing set has 11, 316 objects.

The task is, given a set of 3 images of a single object, induce a density model over images of that object. This is a very challenging problem because the target image camera is arbitrary and unknown, and the background may also change dramatically. Some products are shown cleanly with a white background, and others are shown in a usage context. Some views show the entire product, and others zoom in on a small region.

For this dataset, we found it important to use a multiscale architecture as in Reed et al. (2017). We used three scales: $8 \times 8$, $16 \times 16$ and $32 \times 32$. The base scale uses the standard PixelCNN architecture with 12 layers and 128 planes per layer, with 512 planes in the penultimate layer. The

upscaling networks use 18 layers with 128 planes each. In Attention PixelCNN, the second half of the layers condition on attention features in both the base and upscaling networks.

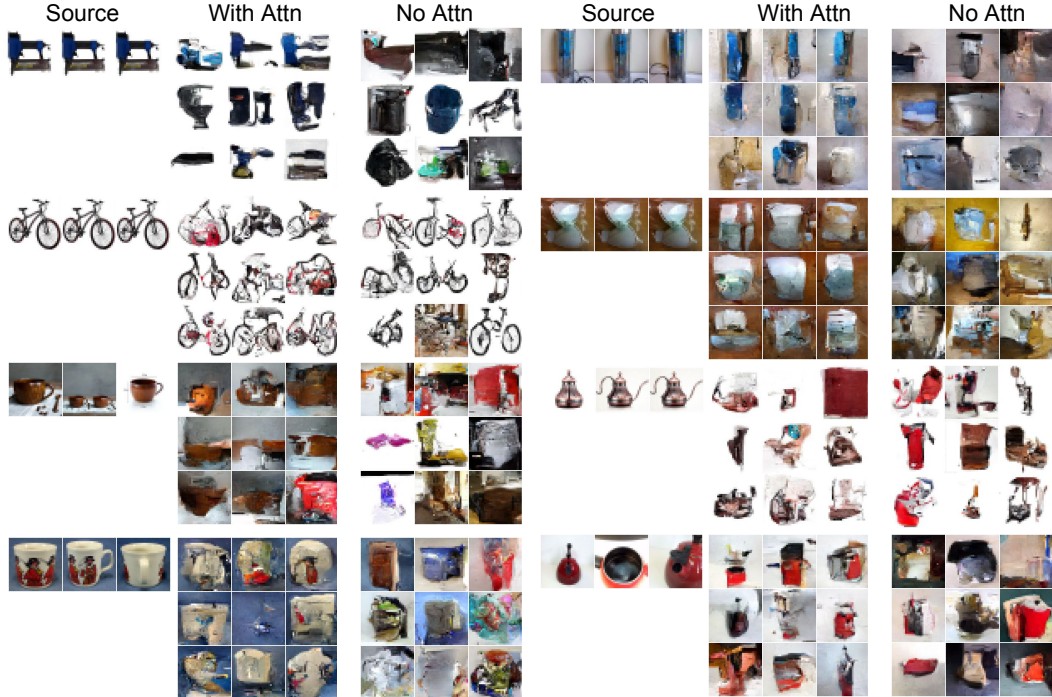

Figure 5: Stanford online products. Samples from Attention PixelCNN tend to match textures and colors from the support set, which is less apparent in samples from the non-attentive model.

Figure 5 shows the result of sampling with the baseline PixelCNN and the attention model. Note that in cases where fewer than 3 images are available, we simply duplicate other support images.

We observe that the baseline model can sometimes generate images of the right broad category, such as bicycles. However, it usually fails to learn the style and texture of the support images. The attention model is able to more accurately capture the objects, in some cases starting to copy textures such as the red character depicted on a white mug.

Interestingly, unlike the other datasets we do not observe a quantitative benefit in terms of test likelihood from the attention model. The baseline model and the attention model achieve 2.15 and 2.14 nats/dim on the validation set, respectively. While likelihood appears to be a useful objective and when combined with attention can generate compelling samples, this suggests that other quantitative criterion besides likelihood may be needed for evaluating few-shot visual concept learning.

## 5 CONCLUSIONS

In this paper we adapted PixelCNN to the task of few-shot density estimation. Comparing to several strong baselines, we showed that Attention PixelCNN achieves state-of-the-art results on Omniglot and also promising results on natural images. The model is very simple and fast to train. By looking at the attention weights, we see that it learns sensible algorithms for generation tasks such as image mirroring and handwritten character drawing. In the Meta PixelCNN model, we also showed that recently proposed methods for gradient-based meta learning can also be used for few-shot density estimation, and also achieve state-of-the-art results in terms of likelihood on Omniglot.

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

# 6 APPENDIX

## 6.1 ADDITIONAL SAMPLES

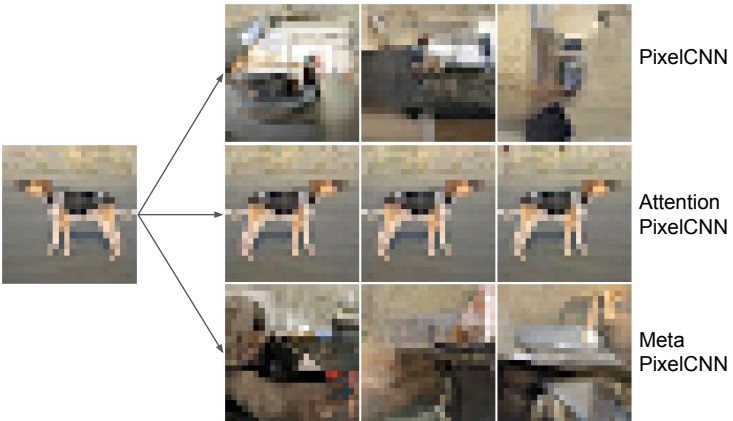

Figure 6: Flipping $24 \times 24$ images, comparing global-conditional, attention-conditional and gradient-conditional (i.e. MAML) PixelCNN.

## 6.2 QUALITATIVE COMPARISON TO CONVDRAW

Although all PixelCNN variants outperform the previous state-of-the-art in terms of likelihood, prior methods can still produce high quality samples, in some cases clearly better than the PixelCNN samples. Of course, there are other important factors in choosing a model that may favor autoregressive models, such as training time and scalability to few-shot density modeling on natural images. Also, the Attention PixelCNN has only 286K parameters, compared to 53M for the ConvDRAW. Still, it is notable that likelihood and sample quality lead to conflicting rankings of several models.

The conditional ConvDraw model used for these experiments is a modification of the models introduced in (Gregor et al., 2015; Rezende et al., 2016), where the support set images are first encoded with 4 convolution layers without any attention mechanism and then are concatenated to the ConvLSTM state at every Draw step (we used 12 Draw-steps for this paper). The model was trained using the same protocol used for the PixelCNN experiments.

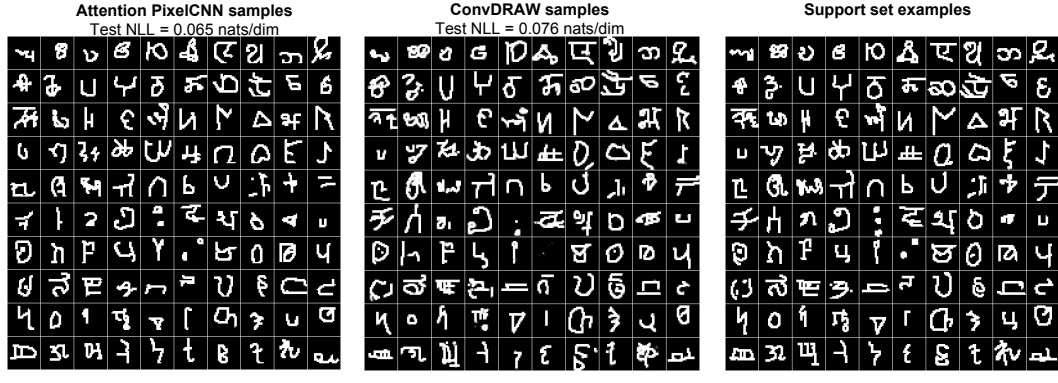

Figure 7: Comparison to ConvDRAW in 4-shot learning.

