# OpenReview forum: "Few-shot Autoregressive Density Estimation: Towards Learning to Learn Distributions"
_ICLR.cc/2018/Conference — Accept (Poster)_

### Official Review · AnonReviewer2 · 2017-11-27
**A solid paper**

**Rating:** 7
**Confidence:** 4

**Review:**

This paper considers the problem of one/few-shot density estimation, using metalearning techniques that have been applied to one/few-shot supervised learning. The application is an obvious target for research and some relevant citations are missing, e.g. "Towards a Neural Statistician" (Edwards et al., ICLR 2017). Nonetheless, I think the current paper seems interesting enough to merit publication.

The paper is well-produced, i.e. the overall writing, visuals, and narrative flow are good. It was easy to read the paper straight through while understanding both the technical details and more intuitive motivations.

I have some concerns about the architectures and experiments presented in the paper. For architectures: the attention-based model seems powerful but difficult to scale to problems with more inputs for conditioning, and the meta PixelCNN model is a standard PixelCNN trained with the MAML approach by Finn et al. For experiments: the ImageNet flipping task is clearly tailored to the strengths of the attention-based model, and the presentation of the general Omniglot results could be improved. The image flipping experiment is neat, but the attention-based model's strong performance is unsurprising. I think the results in Tables 1/2 should be merged into a single table. It would make it clear that the MAML-based and attention-based models achieve similar performance on this task.

Overall, I think the paper makes a nice contribution. The paper could be improved significantly, e.g., by showing how to scale the attention-based architecture to problems with more data or by designing an architecture specifically for use with MAML-based inference.

---

### Official Review · AnonReviewer3 · 2017-11-28
**Few shot learning with autoregressive density estimation**

**Rating:** 6
**Confidence:** 4

**Review:**

This paper focuses on few shot learning with autoregressive density estimation. Specifically, the paper improves PixelCNN with  1) neural attention, 2) meta learning techniques, and shows that the model achieve STOA few showt density estimation on the Omniglot dataset and demonstrate the few showt image generation on the Stanford Online Products dataset.

The model is interesting, however, several details are not clear, which  makes it harder to repeat the model and the experimental results. For example, what is the reason to use the (key, value) pair to encode these support images, what does the "key" means and what is the difference between "keys" and "values"? In the experiments, the author did not explain the meaning of "nats/dim" and how to compute it. Another question is about the repetition of the experimental results. We know that PixelCNN is already a quite complicated model, it would be even harder to implement the proposed model. I wonder whether the author will release the official code to public to help the community?

---

### Official Review · AnonReviewer1 · 2017-11-29

**Rating:** 6
**Confidence:** 5

**Review:**

This paper focuses on the density estimation when the amount of data available for training is low. The main idea is that a meta-learning model must be learnt, which learns to generate novel density distributions by learn to adapt a basic model on few new samples. The paper presents two independent method.

The first method is effectively a PixelCNN combined with an attention module. Specifically, the support set is convolved to generate two sets of feature maps, the so called "key" and the "value" feature maps. The key feature map is used from the model to compute the attention in particular regions in the support images to generate the pixels for the new "target" image. The value feature maps are used to copmpute the local encoding, which is used to generate the respective pixels for the new target image, taking into account also the attention values. The second method is simpler, and very similar to fine-tuning the basis network on the few new samples provided during training. Despite some interesting elements, the paper has problems.

First, the novelty is rather limited. The first method seems to be slightly more novel, although it is unclear whether the contribution by combining different models is significant. The second method is too similar to fine-tuning: although the authors claim that \mathcal{L}_inner can be any function that minimizes the total loss \mathcal{L}, in the end it is clear that the log-likelihood is used. How is this approach (much) different from standard fine-tuning, since the quantity P(x; \theta') is anyways unknown and cannot be "trained" to be maximized.

Besides the limited novelty, the submission leaves several parts unclear. First, why are the convolutional features of the support set in the first methods divided into "key" and "value" feature maps as in p_key=p[:, 0:P], p_value=p[:, P:2*P]? Is this division arbitrary, or is there a more basic reason? Also, is there any different between key and value? Why not use the same feature map for computing the attention and computing eq (7)?

Also, in the first model it is suggested that an additional feature can be having a 1-of-K channel for the supporting image label: the reason is that you might have multiple views of objects, and knowing which view contributes to the attention can help learning the density. However, this assumes that the views are ordered, namely that the recording stage has a very particular format. Isn't this a bit unrealistic, given the proposed setup anyways?

Regarding the second method, it is not clear why leaving this room for flexibility (by allowing L_inner to be any function) to the model is a good idea. Isn't this effectively opening the doors to massive overfitting? Besides, isn't the statement that the function \mathcal{L}_inner void? At the end of the day one can also claim the same for gradient descent: you don't need to have the true gradients of the true loss, as long as the objective function obtains gradually lower and lower values?

Last, it is unclear what is the connection between the first and the second model. Are these two independent models that solve the same problem? Or are they connected?

Regarding the evaluation of the models, the nature of the task makes the evaluation hard: for real data like images one cannot know the true distribution of particular support examples. Surrogate tasks are explored, first image flipping, then likelihood estimation of Omniglot characters, then image generation. Image flipping does not sound a very relevant task  to density estimation, given that the task is deterministic. Perhaps, what would make more sense would be to generate a new image given that the support set has images of a particular orientation, meaning that the model must learn how to learn densities from arbitrary rotations. Regarding Omniglot character generation, the surrogate task of computing likelihood of known samples gives a bit better, however, this is to be expected when combining a model without attention, with an attention module.

All in all, the paper has some interesting ideas. I encourage the authors to work more on their submission and think of a better evaluation and resubmit.

---

### Public Comment · (anonymous) · 2017-11-09
**Previous work**

Hi I think it would be worth adding https://arxiv.org/abs/1612.02192 to your related work.

---

### Author Response · Authors · 2017-12-26
**Clarifications about the method, and corrections of two important factual errors in a review.**

We sincerely thank all reviewers for their thoughtful feedback. We note that AR2 and AR3 both recommend accepting the paper. AR1 recommends reject, although we believe there are a few critical factual errors in that review that, once corrected, should be reflected in a higher score.

Below we respond to each review:

AR1:

We think there are a few important factual errors in this review regarding our method, which should have a substantial effect on the review score. We address these below, and will attempt to improve our writing in the paper to make these points clearer.

Fine-tuning: Meta PixelCNN inference is in fact different than standard fine-tuning by gradient descent. With traditional fine-tuning, the procedure is ad-hoc (e.g. how many fine-tuning gradient steps, what learning rate, what batch size) and needs to be carefully designed to avoid under- or over-fitting. With Meta PixelCNN (and model-agnostic meta learning approaches in general), the critical difference is that the fine-tuning process itself is learned. The key reference that will further clarify this point is https://arxiv.org/abs/1703.03400. https://arxiv.org/abs/1710.11622 provides further theoretical justification.

Inner loss: In fact L_{inner} is learned; we do not use likelihood as the inner loss. So we indeed learn to maximize likelihood without computing likelihoods at test time, as claimed in the paper.

Below we respond to the rest of the review feedback.

Clarity regarding contribution of different model aspects: For the first method (Attention PixelCNN), we demonstrate a clear quantitative benefit of adding attention to the baseline PixelCNN. Although (Attention + Autoregressive Image Model) is a natural idea, we prove that it does indeed work and show a simple and effective implementation, which will be valuable to the research community.

Why use separate key and value? As you suggest it is possible to use the same vector as both key and value. However, separating them may give the network greater flexibility. An ablation here where key and value are the same could be a good experiment, which we are happy to add to the paper.

Assumption of ordered support set: The order can be randomly chosen (and in fact is in our experiments), so the use of a channel for support image identifier should not limit the generality of the method.

Why flexibility of L_{inner} is useful: There are several reasons that we might want L_{inner} to be flexible. For example, a learned L_{inner} may be more efficient to compute than alternatives, as in this paper, or L_{inner} may require less supervision, for example see https://arxiv.org/abs/1709.04905.

Connection between first and second model: The only connection is that they are autoregressive models based on PixelCNN. They are independent models.

AR2:

Presentation: Thank you for the suggestions on how to improve the presentation; indeed combining the tables seems like a good idea.

Scalability of attention: Indeed, this is one of the major challenges in scaling to high-resolution images. Potentially the memory would need to become hierarchical, or we would need to delve more into multiscale variations of the few-shot learning model, which is an interesting area of future research.

AR3:

Meaning of keys/values: The pairs of (query, key) vectors are used to compute the attention scores. Then, the “read” output of memory is the sum of all value vectors in memory each weighted by the normalized attention scores.

Log-likelihood results units: “Bits/dim” results are interpretable as the number of bits that a compression scheme based on the PixelCNN model would need to compress every RGB color value (see e.g. https://arxiv.org/pdf/1601.06759.pdf page 6 for discussion). Nats/dim is the same but multiplied by ln(2). Concretely, in TensorFlow we can compute this value using tf.softmax_cross_entropy_with_logits or tf.sigmoid_cross_entropy_with_logits and then dividing by the total number of dimensions in the image.

Public PixelCNN replication: A great resource for this is https://github.com/openai/pixel-cnn, which is state of the art, and straightforward to modify. Furthermore, we are happy to help guide researchers replicate our experiments, especially on Omniglot which is now a common benchmark.

---

### Decision · Program_Chairs · 2018-01-29
**ICLR 2018 Conference Acceptance Decision**

**Decision:**

Accept (Poster)

**Comment:**

This paper incorporates attention in the PixelCNN model and shows how to use MAML to enable few-shot density estimation. The paper received mixed reviews (7,6,4). After rebuttal the first reviewer updated the score to accept. The AC shares the concern of novelty with the first reviewer. However, it is also not trivial to incorporate attention and MAML in PixelCNN, thus the AC decided to accept the paper.